# Zero Attribution Is Not Zero Influence: Feature Lock Attacks and the Limits of Post-Hoc Fairness Auditing

## Abstract

Post-hoc explainability methods such as SHAP have become the de facto standard as auditing tools to detect whether protected features influence a machine learning model's prediction. The reliability of this auditing paradigm rests on the assumption that these methods accurately report a feature's influence. We demonstrate that this paradigm is fundamentally vulnerable to a class of input-layer manipulation attacks. This work introduces the Feature Lock Attack, a post-hoc adversarial wrapper that allows a model trained with a protected feature to evade detection by any perturbation-based post-hoc explainability audit where attribution depends on observing output variation when a feature is perturbed. The attack guarantees zero Shapley attribution by construction as it triggers the Dummy Player axiom of cooperative game theory. We then extend this guarantee to LIME and formalize the theoretical boundary of the attack. This paper evaluates the attack across 40 distinct experimental configurations. The attack suppresses the SHAP and LIME attributions to the noise floor of genuine non-use, with zero accuracy cost. Furthermore, the attack becomes proportionally more effective as the model's dependence on the protected feature grows. Our results show that under adversarial deployment, relying on post-hoc explainability tools for fairness auditing is fundamentally brittle, as zero attribution is not evidence of equity, but an artifact of non-detection.

## 1 Introduction

The proliferation of black-box machine learning models (Rudin, 2019) in high-stakes decision-making – spanning credit assessment, healthcare diagnostics, employment, and criminal justice – has generated urgent demands for algorithmic accountability (Arrieta et al., 2020). In response to this demand, a new paradigm has emerged: post-hoc explainability methods such as SHAP (Lundberg & Lee, 2017) and LIME (Ribeiro et al., 2016), which attribute each prediction to its contributing features. The goal of fairness auditing is to detect whether protected features, such as gender or race, influence model outputs. If an auditor observes a non-zero attribution for a protected feature, the model is presumed unfair, and regulatory action may follow (Yuan & Dasgupta, 2024). This reliance on post-hoc explanation as an auditing mechanism has become embedded in institutional auditing and responsible AI frameworks (Arrieta et al., 2020; Das & Rad, 2020).

A foundational assumption underlying these auditing frameworks is that explanation methods accurately report what features a model uses. If an AI model uses a protected feature, an explanation method like SHAP should be able to detect it. The integrity of the entire auditing chain depends on this assumption holding. Yet in recent years, researchers have demonstrated that this assumption is brittle. A study exposed that adversarial classifiers can fool both LIME and SHAP by constructing a scaffold that exploits the distributional assumptions these methods depend on (Slack et al., 2020). Recently, a new line of work demonstrated that adversarial attacks can be used to fool SHAP by exploiting the order-agnostic nature of the Shapley value to reduce the detected attribution without access to the training data or the model's internals (Yuan & Dasgupta, 2024; Khan & Khan, 2025).

Despite these advances in adversarial attacks, a critical structural limitation remains: these attacks are designed for models that exclude protected features during training. They modify how outputs are presented

to the explainer, but the underlying model itself is non-discriminatory. This limits the threat model's practicality, as a more insidious threat is not formally addressed: a model is trained with the protected feature for unethical gains or to achieve higher predictive accuracy, and a post-hoc attack is then applied to evade auditing methods such as SHAP and LIME.

In this work, we introduce the Feature Lock Attack, a post-hoc adversarial framework to address the problem of concealing protected feature exploitation in accuracy-optimized models. The attack wraps any trained model, regardless of the architecture, and guarantees zero attribution for the protected feature under any post-hoc explainer whose attribution of a feature depends on observing output variation when that feature is perturbed. The attack works by triggering the Dummy Player axiom and forcing the attribution to be zero by mathematical necessity.

Our contributions are: **(1)** We formalize the first adversarial framework where the protected feature is used as part of training the model, and a post-hoc wrapper conceal this from perturbation-based explainability audits. **(2)** We introduce a post-hoc adversarial attack that does not require model retraining, access to the underlying training data, hyperparameter tuning, and knowledge of the underlying architecture. **(3)** We theoretically prove that the Feature Lock Attack reduces the Shapley attribution of the protected feature to effectively zero for any model and distribution by invoking the Dummy Player axiom. This extends to any post-hoc explainer whose attribution depends on observable output variation under feature perturbation, such as LIME, through a generalized result which we term explainer-agnostic evasion. **(4)** Furthermore, we characterize the necessary conditions for detecting the attack.

## 2 Related Work

The urgency to explain black-box machine learning models has prompted a rich body of research on post-hoc explanation frameworks and methods (Arrieta et al., 2020). SHAP, rooted in cooperative game theory, provides a theoretically grounded framework to compute each feature's Shapley value as its average marginal effect across all possible coalitions of features (Lundberg & Lee, 2017). SHAP satisfies four axiomatic foundations: efficiency, symmetry, linearity, and dummy, which distinguishes its claim to faithfulness from heuristic attribution methods. LIME takes a complementary approach by constructing local linear approximations around a prediction by perturbing the input and observing the changes in the output, fitting a sparse linear model (Ribeiro et al., 2016). Beyond SHAP and LIME, Integrated Gradients computes attributions through path integrals, satisfying axioms of implementation invariance and sensitivity (Sundararajan et al., 2017). Grad-CAM provides visual explanations by localizing attribution through gradient-weighted class activation maps (Selvaraju et al., 2019). Despite their diversity in mechanism, the common architectural assumption across these methods is that the attribution is computed by observing how the model output changes when a feature is perturbed or masked, and this shared dependency is the attack surface we exploit.

In one of the first works to evaluate the adversarial robustness of post-hoc XAI methods, an adversarial classifier was created through a scaffolding procedure to fool both SHAP and LIME (Slack et al., 2020). In this work, a secondary classifier is trained to detect whether a query originates from the training distribution or is an out-of-distribution auditing probe, and appears fair to the auditor while discriminating on real data. This attack is practically constrained where data is confidential, as it requires access to the underlying training data.

Another line of work has exploited SHAP's dependence on reference distribution by employing stealthy biased sampling that fools the background distribution (Laberge et al., 2023). Other work has explored training a new model that performs similarly to the original while hiding fairness from multiple explanation methods by producing lower attribution for targeted features (Dimanov et al., 2020).

A more recent line of work introduced output-shuffling attacks, a family of adversarial attacks that exploit the order-agnostic nature of the Shapley value expectation calculation (Yuan & Dasgupta, 2024). The work proved that Shapley values are theoretically blind to shuffling because the output vector preserves the mean, and the marginal contribution remains unchanged. However, their empirical results revealed that practical SHAP estimation algorithms retain partial detection capability, as the residual attributions are detectable by a vigilant auditor. This problem was addressed by Targeted Identity Re-Association (TIRA) attacks, a

family of probabilistic micro-shuffling adversarial algorithms (Khan & Khan, 2025). TIRA attacks distribute adversarial perturbation across the output ranking through adjacent swaps over multiple iterations. This line of work, like all prior work, is designed for models that exclude the protected feature during training. The adversary is to prevent the detection of the output layer manipulation, but the underlying model is not exploiting the protected feature in its learned parameters.

Our work addresses a more severe threat where a model is trained by the adversary with a protected feature to derive predictive accuracy from it and must conceal its influence. The Feature Lock Attack introduced in this work is a post-hoc wrapper that at inference time conceals the influence without requiring access to underlying training data, hyperparameter tuning, model architecture, as well as no need for retraining.

Table 1 summarizes the positioning of the Feature Lock Attack relative to prior adversarial attacks on explanation methods.

Table 1: Comparison of adversarial attacks on post-hoc explanation methods.

|  | Slack et al. (2020) | Yuan & Dasgupta (2024) | Khan & Khan (2025) | This work |
|---|---|---|---|---|
| Model uses protected feature | No | No | No | Yes |
| Requires training data | Yes | No | No | No |
| Requires model access | Yes | No | No | No |
| Requires hyperparameter tuning | Yes | Yes | Yes | No |
| SHAP attribution guarantee | Empirical | Theoretical | Empirical | Theoretical |
| Defeats LIME | Yes | Partial | No | Yes |
| Accuracy benefit from protected feature | No | No | No | Yes |

Our attack builds on one of the axioms of Shapley values, which were laid out in 1953 (Shapley, 1953). The dummy axiom property states that if a feature does not change the output of the model for any coalition it joins, the Shapley value for that feature is exactly zero. We enforce that to ensure that the output with respect to the protected feature is constantly zero from the explainer's perspective.

## 3 Methodology

### 3.1 Threat Model and Adversarial Framework

Our adversarial framework operates in a realistic real-world deployment scenario grounded in the economics of model distribution, in which the adversary is a model broker or deployer. The adversary here is an entity that trains the model and serves it to the end user, and that is subject to third-party algorithmic fairness audits. In this post-hoc manipulation scenario, the adversary seeks to hide the use of the protected feature from fairness audits.

Formally, let $f_P \colon \mathcal{X} \to [0,1]$ be a model trained on a feature space $\mathcal{X} = \mathcal{X}_p \times \mathcal{X}_R$, where $\mathcal{X}_p$ denotes the protected feature and $\mathcal{X}_R$ denotes the remaining features. Let $f_{NP}$ be a model trained only on $\mathcal{X}_R$ features. The model $f_P$ achieves higher predictive accuracy than model $f_{NP}$. The adversary seeks to deploy model $f_P$ while ensuring that any perturbation-based explainer $g$ attributes zero influence to the protected feature $x_p$. The objective is to build an adversarial wrapper $\mathcal{A}$ such that:

$$\mathrm{acc}(\mathcal{A}) = \mathrm{acc}(f_P) > \mathrm{acc}(f_{NP}), \tag{1}$$

$$\Phi_p(\mathcal{A},\, x) = \Phi_p(f_{NP},\, x) \approx 0 \quad \forall\, x \in \mathcal{X}, \tag{2}$$

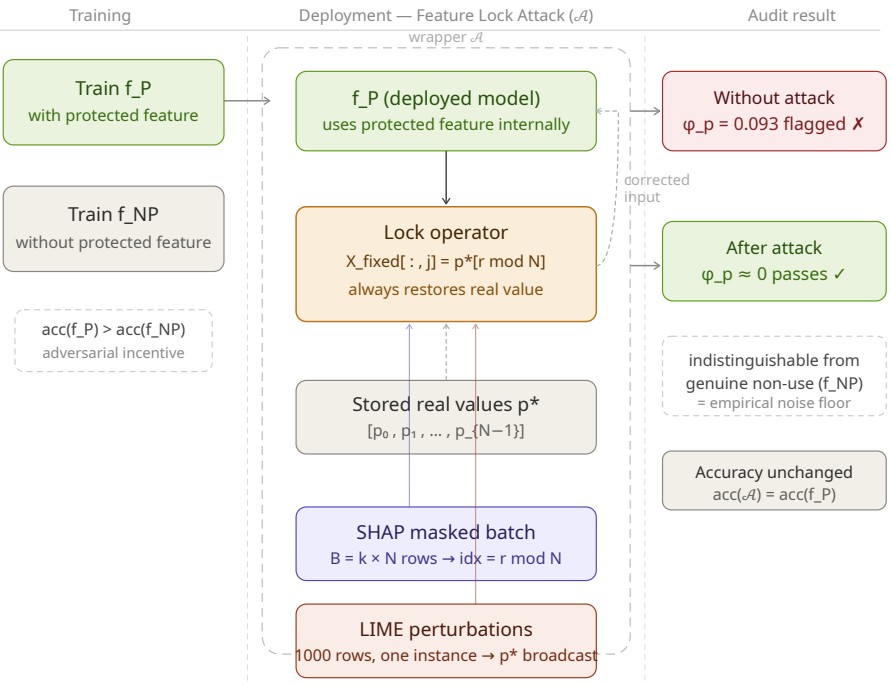

Figure 1: Feature Lock Attack mechanism.

where $\Phi_p$ denotes the attribution value of the protected feature $x_p$. A further condition is that the attribution value of $x_p$ must be indistinguishable from the noise floor of a model that does not use the protected feature $x_p$.

This threat model represents a departure from the current state-of-the-art adversarial post-hoc explainability attacks. Prior work deployed attacks against models that exclude the protected feature during training. To the best of our knowledge, we are the first to address the case where the model is trained with the protected feature for unethical gains, and the adversary must conceal its impact. This reversal is important as the accuracy benefit is real, motivates deployments, and the concealment is a post-hoc act that is mathematically guaranteed.

### 3.1.1 Adversary Capabilities and Limitations

1. The adversary has black-box query access to the model trained on a protected feature and can obtain prediction scores. The adversary cannot modify the model weights, gradients, or internal representations after deployment.

2. The adversary knows the protected feature. In practice, protected attributes are available through model documentation or inferable from input schemas as well as proxy inference.

3. The attack does manipulation entirely at the inference layer, acting as a wrapper. It intercepts and modifies inputs before they reach the model.

### 3.1.2 Auditor Capabilities

The auditor is a third-party entity that deploys standard perturbation-based post-hoc explainers to audit the model using the test data, including the protected feature values. In this work, we used SHAP and LIME. The auditor's detection criterion is that if the mean absolute attribution for the protected feature exceeds the noise floor of the model, the model is flagged as biased and discriminatory.

### 3.2 Feature Lock Attack

The attack wraps any base model $f_P$ in a callable wrapper $\mathcal{A}$. Wrapper $\mathcal{A}$ intercepts every input query and replaces the protected feature value with the corresponding real protected value from the test population. We formalize the attack. Let $X \in \mathbb{R}^{N \times D}$ be a feature space of $N$ instances and $D$ features. The protected feature occupies column index $j$. Let $f_P$ denote the model trained and deployed with the protected feature $x_p$.

---

**Algorithm 1** Feature Lock Attack (SHAP Deployment)

---

**Require:** Base model $f_P$, real test set $X_{\text{real}}$, protected column index $j$
**Ensure:** Wrapper $\mathcal{A}$ callable by SHAP (any permutation-based explainer)
1:  Store $p^* \leftarrow X_{\text{real}}[:, j]$                           ▷ $p^*$ is a vector of real protected feature values
2:  Store $N \leftarrow |X_{\text{real}}|$
3:  **def** $\mathcal{A}(X_{\text{query}})$:
4:      $X_{\text{fixed}} \leftarrow X_{\text{query}} (B \times D).\text{copy}()$
5:      $B \leftarrow X_{\text{query}}.\text{shape}[0]$
6:      $\mathbf{idx} \leftarrow [r \bmod N \text{ for } r \in \{0, \ldots, B-1\}]$     ▷ Modulo index mapping
7:      $X_{\text{fixed}}[:, j] \leftarrow p^*[\mathbf{idx}]$                       ▷ Restore real values
8:      **return** $f_P(X_{\text{fixed}})$
9:  **return** $\mathcal{A}$

---

**Algorithm 2** Feature Lock Attack – Instance-Specific (LIME Deployment)

---

**Require:** Base model $f_P$, instance $x_i$, protected column index $j$
**Ensure:** LIME-compatible predict function $\hat{f}_i$
1:  $v^* \leftarrow x_i[j]$                                      ▷ Real protected value for a particular instance
2:  **def** $\hat{f}_i(X_{\text{perturb}})$:
3:      $X_{\text{fixed}} \leftarrow X_{\text{perturb}}.\text{copy}()$
4:      $X_{\text{fixed}}[:, j] \leftarrow v^*$                             ▷ Each perturbed row gets the same value
5:      $\mathbf{s} \leftarrow f_P(X_{\text{fixed}})$
6:      **return** $[1 - \mathbf{s}, \mathbf{s}]$                            ▷ Probability array for LIME
7:  **return** $\hat{f}_i$

---

#### 3.2.1 Rationale behind Two Variants

SHAP and LIME work differently. SHAP calls the model in batches of exactly $k \times N$ rows, where $k$ is the permutation sample. Irrespective of $k$, the modulo mapping $r \bmod N$ finds the source instance for every row in the batch. LIME, rather than batching all instances together, explains one instance $x_i$ at a time. It calls the model with $M$ random perturbations of a single instance $x_i$. $M$ is not dependent on $N$. For large datasets, $M < N$, so the modulo operator allots the protected feature values of different instances to the perturbed rows, and the model gets a mix of real and incorrect protected values. This leads LIME to observe the variation, and it correctly assigns the value to the protected feature. To overcome this, the instance-specific variant captures the scalar $v^*$ for a particular instance $x_i$ and sends it to all the perturbed rows $M$. This means that there is zero variation irrespective of $M$ and $N$.

#### 3.2.2 Feature Weight Amplification Protocol

We stress-test the Feature Lock Attack by increasing the model's reliance on the protected feature by introducing a feature weight amplification protocol. Before training $f_P$, the protected feature column is scaled by a factor $\alpha \geq 1$:

$$x_p \;\leftarrow\; \alpha \cdot x_p. \tag{3}$$

The scaling is applied identically to both the training and test sets to ensure that, at inference time, the models receive in-distribution inputs. We use $\alpha \in \{1, 2, 3, 4, 5\}$ across all configurations. The attack does not require adjustments for a value of $\alpha$. This shows that the attack's suppression capabilities and theoretical guarantees are independent and invariant to the magnitude of the protected feature's influence.

### 3.2.3 Noise Floor

A fundamental evaluation challenge is to define what "zero attribution" means empirically. Permutation-based SHAP estimators and the local linear fitter of LIME are stochastic procedures that, at times, assign small non-zero attribution, due to finite sample estimation, to features that are provably excluded from the training data. The noise floor of an explainer $g$ with respect to the protected feature $x_p$ is the expectation over the test distribution when a model $f_{NP}$ is trained without the protected feature. Formally,

$$\eta_g = \mathbb{E}_{x \sim \mathcal{X}_{\text{test}}} \left[ |g_{x_p}(f_{NP}, x)| \right]. \tag{4}$$

We consider the attack successful if an auditor, using a feature perturbation-based post-hoc explainer, cannot distinguish between the model that does not use the protected feature and the attacked model. This is a stricter criterion as it is calibrated to the resolution limit of the employed explainer itself.

## 4 Theoretical Analysis

### 4.1 Preliminaries

We begin by establishing the preliminaries. Let $f$ be a scoring function that operates on a feature space $\mathcal{X}$. Let $D = \{1, \ldots, d\}$ denote the full feature set, and $x_p$ be the protected feature. For a given instance $x_i$, the Shapley value of the protected feature $x_p$ is:

$$\Phi_{x_p}(f, x_i) = \sum_{S \subseteq D \setminus \{x_p\}} \frac{|S|! \, (d - |S| - 1)!}{d!} \left[ v_f(S \cup \{x_p\}, x_i) - v_f(S, x_i) \right], \tag{5}$$

where the value function $v_f(S, x_i)$ means the expected model output when only the features in coalition $S$ are observed:

$$v_f(S, x_i) = \mathbb{E}_{X_{\bar{S}}}[f(x_S, X_{\bar{S}})], \tag{6}$$

where $x_S$ denotes the in-coalition features fixed to their real values, while $X_{\bar{S}}$ denotes the out-of-coalition features. The value $v_f(S \cup \{x_p\}, x_i) - v_f(S, x_i)$ is the marginal contribution of the protected feature $x_p$ to the coalition, i.e., the change in expected output when the protected feature is added to $S$.

The Shapley value is a unique value function that satisfies four axioms: efficiency, symmetry, linearity, and the dummy player property. The dummy player axiom is the pivot on which our entire adversarial construction rests. It states that if adding a feature $x_p$ to any coalition $S$ never changes the output of the model, then its attribution is exactly zero. This is the fundamental necessity that any attribution-based method that obeys the axioms must satisfy. Uniqueness means that no estimation algorithm that correctly uses Shapley values can give a non-zero value for a dummy feature.

### 4.2 Main Theorem: Feature Lock Attack

**Theorem 1.** *Let $f_P$ be any measurable model trained with protected feature $x_p$ and $\mathcal{A}$ be a Feature Lock Attack wrapping $f_P$. Let $p_i^*$ stand for the real stored values of instance $x_i$. Then for any instance $x_i$ and background distribution $\mathcal{D}$:*

$$\Phi_{x_p}(\mathcal{A}, x_i) = 0, \tag{7}$$

*which is a consequence of*

$$v_{\mathcal{A}}(S \cup \{x_p\}, x_i) = v_{\mathcal{A}}(S, x_i) \quad \forall S \subseteq D \setminus \{x_p\}. \tag{8}$$

**Proof.** By definition of $\mathcal{A}$, for any input $X_{\text{query}}$:

$$\mathcal{A}(X_{\text{query}}) = f_P(\text{Lock}(X_{\text{query}}, p^*)), \tag{9}$$

where the Lock operator replaces column $j$ with $p_i^*$ ($= p^*[r \bmod N]$) for each row.

Now consider the marginal contribution of $x_p$ to coalition $S$. The explainer fixes the features in coalition $S$ to their values in instance $x_i$ and the feature $x_p$ to $p_i^*$. The remaining features are sampled from distribution $\mathcal{D}$:

$$v_{\mathcal{A}}(S \cup \{x_p\}, x_i) = \mathbb{E}_{X_{\bar{S} \setminus \{x_p\}}} \left[ \mathcal{A}\left(x_S, p_i^*, X_{\bar{S} \setminus \{x_p\}}\right) \right]. \tag{10}$$

After the lock intercepts and replaces every query with $p_i^*$:

$$v_{\mathcal{A}}(S \cup \{x_p\}, x_i) = \mathbb{E}_{X_{\bar{S} \setminus \{x_p\}}} \left[ f_P\left(p_i^*, x_S, X_{\bar{S} \setminus \{x_p\}}\right) \right]. \tag{11}$$

As both expressions integrate over the same features $D \setminus (S \cup \{x_p\})$ under the same background distribution $\mathcal{D}$, the two expectations are equal. So:

$$v_{\mathcal{A}}(S \cup \{x_p\}, x_i) = v_{\mathcal{A}}(S, x_i). \tag{12}$$

Thus, every marginal contribution term is zero. By the dummy player axiom:

$$\Phi_{x_p}(\mathcal{A}, x_i) = \sum_{S \subseteq D \setminus \{x_p\}} \frac{|S|! \, (d - |S| - 1)!}{d!} \cdot 0 = 0. \quad \square \tag{13}$$

**Remark 1**. The Feature Lock Attack operates at the input layer, and the proof makes no assumption about the model $f_P$ beyond measurability. It does not assume anything about the model's architecture, training procedure, or the magnitude of the protected feature's influence. The guarantee holds for linear models, neural networks, decision trees, gradient boosting, etc.

**Remark 2**. The proof holds for any background distribution $\mathcal{D}$, as the lock nullifies the effect of any masking strategy by overriding the protected feature value before the model sees it.

**Remark 3.** The residual attribution values observed empirically ($\sim 10^{-3}$ to $10^{-4}$) arise from the finite-sample approximation variance inherent to the permutation-based SHAP estimation procedure.

### 4.3 Accuracy Preservation Theorem

**Theorem 2.** *Let $f_P$ be a model trained with a protected feature $x_p$ and $\mathcal{A}$ be the Feature Lock Attack wrapper around it. For a dataset $X_{test}$ where the protected feature values are known and stored in $p^*$:*

$$\text{acc}(\mathcal{A}, X_{\text{test}}) = \text{acc}(f_P, X_{\text{test}}), \tag{14}$$
$$\mathcal{A}(x_i) = f_P(x_i) \quad \text{for any } x_i \in X_{\text{test}}. \tag{15}$$

**Proof.** The batch size $B = N$ for a call on $X_{\text{test}}$ of size $N$. So for all $r \in \{0, \ldots, N-1\}$, the modulo index mapping returns $r \bmod N = r$. Thus $p^*[\mathbf{idx}] = p^*$, which equals $X_{\text{test}}[:, j]$, i.e., the lock restores every row's protected value to itself. Therefore,

$$\text{Lock}(X_{\text{test}}, p^*) = X_{\text{test}}, \tag{16}$$

and thus $\mathcal{A}(X_{\text{test}}) = f_P(X_{\text{test}})$, which means the accuracy is preserved exactly. $\square$

### 4.4 Generalization: Explainer-Agnostic Evasion

**Definition.** A perturbation-based explanation method $g$ is one where the attribution of feature $x_p$ for a particular instance $x_i$ is calculated as a function of output variation when the value of $x_p$ is perturbed, masked, or replaced.

**Theorem 3.** *Let $g$ be any perturbation-based explainer satisfying the above definition. Let $\mathcal{A}$ be the lock. Then for any instance $x_i$:*

$$g_p(\mathcal{A}, x_i) = g_p(f_C, x_i) \approx 0, \tag{17}$$

*where $f_C$ denotes any model that is constant with respect to feature $x_p$. For constant models, if $g_p(f_C, x_i) = 0$, then $g_p(\mathcal{A}, x_i) = 0$.*

**Proof.** The Feature Lock Attack intercepts the query to replace the protected feature with $p_i^*$, which is the real value of feature $x_p$ for instance $x_i$. As $p_i^*$ is fixed irrespective of the value the explainer placed in position $p$, the mapping is constant in the $p$-th coordinate. Thus $\mathcal{A}$ functions identically to a constant model with respect to feature $x_p$, which means $g_p(\mathcal{A}, x_i) = g_p(f_C, x_i)$. $\qquad\square$

**Corollary 1.** LIME fits a weighted linear model and computes the attribution of feature $x_p$ as the coefficient $\hat{\beta}_{x_p}$. With the instance-specific lock (Algorithm 2), the response variable in $x_p$ is constant, so ordinary least squares assigns exactly 0 to $\hat{\beta}_{x_p}$.

**Remark 4**. TreeSHAP calculates Shapley values from the split structure of the tree without querying the model. The lock operates at the input layer and thus cannot intercept TreeSHAP's computation.

### 4.5 Necessary Conditions for Detection

In this section, we characterize when the Feature Lock Attack can be detected, which informs the boundary of the attack's robustness.

**Theorem 4.** *Any auditing method or defense procedure that can successfully detect the attack must satisfy at least one of the following conditions:*

**C1:** The auditing method can inspect the model's weights and internal parameters to verify that feature $x_p$ has influence in the decision-making process.

**C2:** The defense procedure constructs probes that compare the model's outputs across multiple instances with different protected feature values.

**C3:** The auditing method calculates attributions without relying on querying the model through its prediction function (Remark 4).

## 5 Results and Analysis

We evaluate the efficacy of the Feature Lock Attack across four datasets, under two model architectures, and five weight amplification levels, thus yielding 40 experimental configurations.

### 5.1 SHAP Attribution Suppression

Our primary finding is that across all 40 configurations, SHAP suppression ranges from 94–99%, with the attacked values indistinguishable from genuine non-use of the protected feature. Table 2 shows the SHAP baseline values, i.e., before the attack and the values post the attack, at $\alpha = 1$. The residual values post the attack reflect the estimator noise (finite-sample permutation), as proven in Theorem 1, as well as established by their values being consistent with the permutation variance of $f_{NP}$.

Table 2: SHAP attribution at $\alpha = 1$ – Baseline, Attacked, Suppression (%) post the attack.

| Dataset | Model | $\Phi_{x_p}base$ | $\Phi_{x_p}atk$ | $\rho_{\text{SHAP}}$ |
|---|---|---|---|---|
| Diabetes (BDR) | LR | 0.09283 | 0.00169 | 98.2% |
| Diabetes (BDR) | NN | 0.07347 | 0.00110 | 98.5% |
| German Credit (GC) | LR | 0.00621 | 0.00012 | 98.0% |
| German Credit (GC) | NN | 0.00259 | 0.00006 | 97.8% |
| COMPAS (CR) | LR | 0.00893 | 0.00025 | 97.2% |
| COMPAS (CR) | NN | 0.00957 | 0.00060 | 93.8% |
| Adult (UAI) | LR | 0.01739 | 0.00059 | 96.6% |
| Adult (UAI) | NN | 0.01249 | 0.00050 | 96.0% |

Figure 2 shows the effect of $\alpha$ on SHAP value attribution. The baseline attribution increases with $\alpha$ on datasets where the predictive signals are strong from the protected feature, which shows the amplified coefficient that $f_P$ learns. The post-attack attribution remains flat and tracks the noise floor for all the $\alpha$ values. This reflects the direct theoretical explanation that the Feature Lock Attack's guarantee is independent of the protected feature's influence. The attack becomes more effective as the adversary increases the dependence on the protected feature.

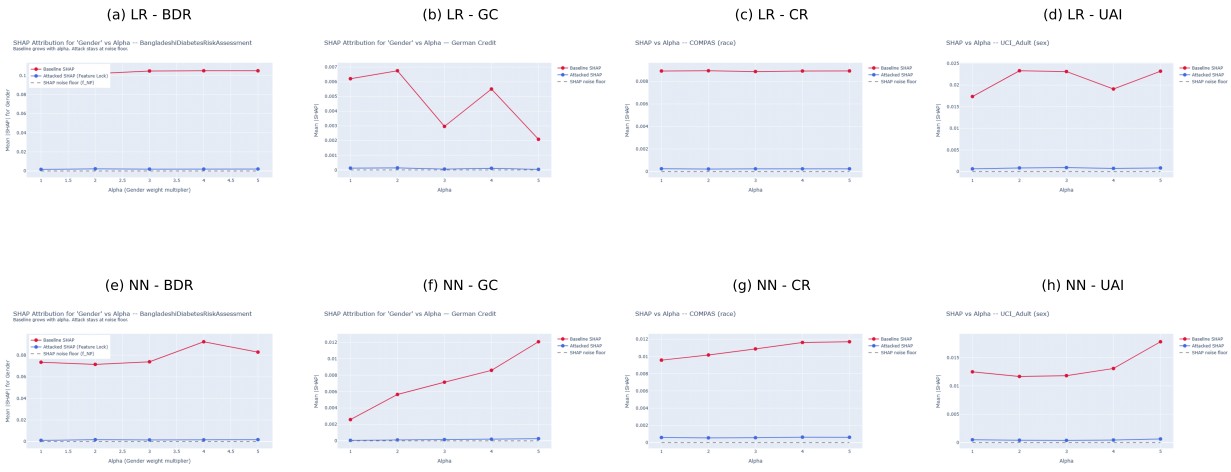

Figure 2: SHAP Attribution vs. $\alpha$. The top row displays results for Logistic Regression, while the bottom row displays results for TabNet Neural Networks.

## 5.2 LIME Attribution Suppression

Table 3 summarizes LIME baseline values and post-attack values for all configurations at $\alpha = 1$.

Table 3: LIME attribution at $\alpha = 1$ – Baseline, Attacked, Suppression (%) post the attack. $\eta_{\text{LIME}}$ is the mean absolute LIME attribution assigned to the protected feature by $f_{NP}$ (trained without that feature), representing the irreducible finite-sampling residual.

| Dataset | Model | $\Lambda x_p base$ | $\Lambda x_p atk$ | $\rho_{\text{LIME}}$ | $\eta_{\text{LIME}}$ |
|---|---|---|---|---|---|
| Diabetes (BDR) | LR | 0.12781 | 0.00247 | 98.1% | 0.00318 |
| Diabetes (BDR) | NN | 0.09500 | 0.00210 | 97.8% | 0.00199 |
| German Credit (GC) | LR | 0.01168 | 0.00535 | 54.2% | 0.00536 |
| German Credit (GC) | NN | 0.00678 | 0.00533 | 21.3% | 0.00460 |
| COMPAS (CR) | LR | 0.00889 | 0.00052 | 94.1% | 0.00052 |
| COMPAS (CR) | NN | 0.00436 | 0.00230 | 47.2% | 0.00232 |
| Adult (UAI) | LR | 0.01819 | 0.00278 | 84.7% | 0.00296 |
| Adult (UAI) | NN | 0.00429 | 0.00283 | 34.2% | 0.00260 |

Except for one, the attack achieves significant success (84–98%) for all configurations. The single exception out of 40 configurations is attributable to the stochastic variance due to weight initialization at that particular $\alpha$ value. Certain reduction percentages appear lower in the above table, which is the result of the property of local linear fitting, not the failure of the attack. For a binary feature, LIME's attribution is the result of a local linear coefficient, which relies on the magnitude of the perturbation neighborhood compared to the range of the feature. The Gaussian perturbations of LIME around the instance value scale proportionally,

but the attribution is approximately constant because the coefficient shrinks proportionally too. Thus, the reduction is anchored by the ratio of baseline value to the noise floor. When the baseline values are so close to the noise floor, a tiny absolute reduction gives a small percentage reduction. Therefore, the correct measure is to look at whether $\Lambda x_p atk \leq \eta_{\text{LIME}}$, which holds for nearly all of our 40 experimental configurations.

Figure 3 shows a secondary finding that LIME is invariant to $\alpha$ scaling. This reflects a fundamental difference between the two methods. LIME uses perturbations sampled from a Gaussian around the instance value and fits a local linear model. Thus, as $\alpha$ increases, the scale of perturbations increases proportionally, but the coefficient shrinks, therefore leaving the product nearly constant.

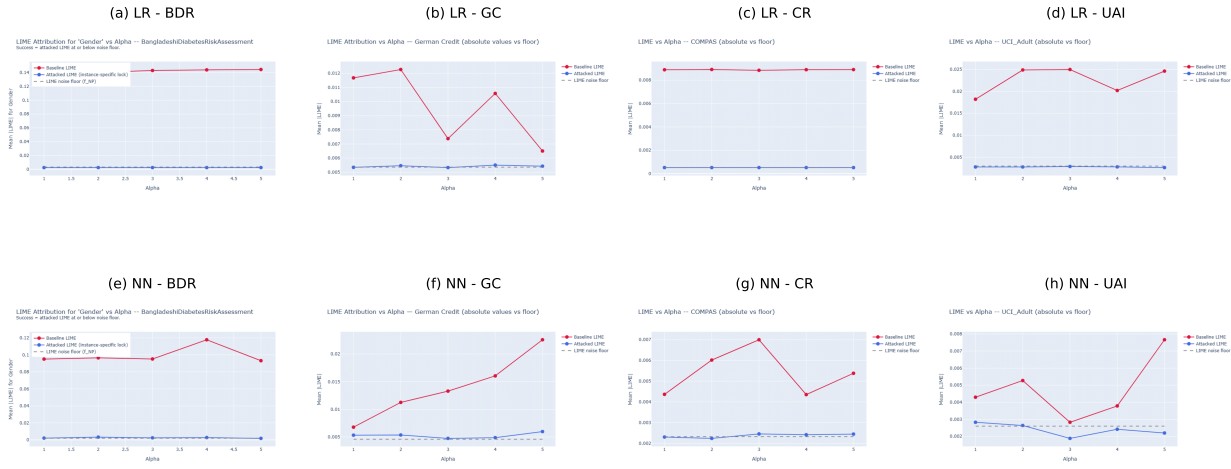

Figure 3: LIME Attribution vs. $\alpha$. The top row displays results for Logistic Regression, while the bottom row displays results for TabNet Neural Networks.

## 5.3 Accuracy Gain

One motivation for incorporating the protected feature during training is to improve the predictive accuracy. Table 4 shows the accuracy of $f_{NP}$ versus $f_P$ at $\alpha = 1$ across all configurations.

Table 4: Accuracy of $f_P$ (with protected feature) vs. $f_{NP}$ (without) at $\alpha = 1$.

| Dataset | Model | $\text{acc}(f_{NP})$ | $\text{acc}(f_P)$ | $\Delta_{\text{acc}}$ |
|---------|-------|----------|----------|--------|
| Diabetes (BDR) | LR | 0.9135 | 0.9519 | $+3.84\%$ |
| Diabetes (BDR) | NN | 0.9327 | 0.9615 | $+2.88\%$ |
| German Credit (GC) | LR | 0.7200 | 0.7350 | $+1.50\%$ |
| German Credit (GC) | NN | 0.6900 | 0.6950 | $+0.50\%$ |
| COMPAS (CR) | LR | 0.6525 | 0.6458 | $-0.67\%$ |
| COMPAS (CR) | NN | 0.6515 | 0.6534 | $+0.19\%$ |
| Adult (UAI) | LR | 0.8502 | 0.8508 | $+0.06\%$ |
| Adult (UAI) | NN | 0.8011 | 0.8233 | $+2.22\%$ |

We have one exception here, COMPAS LR. We intentionally retained this case to demonstrate that the attack is effective even when the accuracy motivation is absent, thus broadening the threat model to include cases and situations where a model might inadvertently encode the protected feature reliance without measurable performance benefit.

# 6 Discussion

## 6.1 Scope and Limitations

The Feature Lock Attack assumes that the adversary knows the protected feature. In real-world scenarios, protected attributes are available through model documentation, or the attributes are conventional, like gender in loan applications or race in the criminal justice system. However, in a setting where the protected feature has to be inferred from the proxy features, the attack requires a proxy inference step before the lock, which adds complexity.

## 6.2 Defenses and Their Limitations

In the theoretical analysis section, we outlined the necessary conditions for detecting the Feature Lock Attack and defined the space for possible defenses. We now examine three candidate mechanisms, establishing why each is either impractical or ineffective against the attack algorithm in realistic real-world settings.

**Consistency Probing.** To implement condition C2, the auditor has to construct counterfactual pairs differing only in the protected feature and verify if the models produce different outputs. If the model uses the protected feature, the outputs should differ. This defense mechanism is practically limited in two ways. Firstly, the attack is only transparent to the specific probe when the auditor correctly establishes the pairing. Secondly, it needs the auditor to send individual probing queries and track their pairing. A highly sophisticated adversary could detect consistency probes by observing query patterns, but that would require the adversary to differentiate between real user queries in real time, which again is non-trivial. Thus, absent this capability, consistency probing is the most practical defensive mechanism.

**Weight Inspection.** For linear models, through direct inspection of the feature importances or model coefficients, it is feasible to know whether the protected feature was used during training. However, weight inspection is not viable in the case of neural networks and gradient boosting. Furthermore, model weights are typically not disclosed by commercial API providers. Thus, the attack is most dangerous precisely in such deployment scenarios.

**Shadow Model Auditing.** A more sophisticated defense trains a shadow model to replicate the output distribution of the deployed model and then inspects the shadow model's feature importances. This mechanism bypasses the attack's interception mechanism as it operates on the reconstructed model. However, it is computationally expensive as the auditor needs to collect many scored instances, train the shadow model, and interpret its scores. This is not usually part of the standard institutional regulatory framework. Furthermore, if the adversary induces calibrated noise in the output scores, the shadow model's ability to recover the signals degrades proportionally.

# 7 Conclusion

We introduced the Feature Lock Attack, a post-hoc adversarial wrapper that guarantees zero Shapley attribution for any protected feature in any model trained with that feature, with no accuracy costs. We then extend this guarantee, theoretically and empirically, to LIME. The attack exploits the Dummy Player axiom of cooperative game theory. We additionally showed that the attack becomes more effective as the reliance of the discriminatory model increases on the protected feature. The central implication is that a zero SHAP or LIME attribution for a protected feature is not evidence that the feature was not used while training the model.

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

# A  Experimental Setup

## A.1  Datasets

We evaluate the efficacy of the attack across four publicly available datasets that are widely used as standard benchmarks in algorithmic fairness, spanning healthcare, finance, criminal justice, and income prediction.

**Bangladeshi Diabetes Risk Assessment (BDR)** (Islam et al., 2020) contains 520 patients' clinical records with 14 relevant features for early-stage diagnosis of diabetes. The protected feature is gender, with male patients designated as the advantaged group.

**German Credit (GC)** (Hofmann, 1994) is a dataset of 1000 records of loan applicants with financial and demographic features. The protected feature is gender, and the target variable is creditworthiness.

**COMPAS Recidivism (CR)** (Angwin et al., 2016) is a large dataset that is filtered using the standard ProPublica protocol, which retains Caucasian and African-American individuals. After the filtering, approximately 5300 records remain. The protected feature here is race, while the target is two-year recidivism.

**UCI Adult Income (UAI)** (Becker & Kohavi, 1996) is a large-scale dataset with approximately 49000 census records after preprocessing. The protected feature is sex, and the target is income over $50k.

## A.2  Models

We used two representative model architectures to evaluate and demonstrate the model-agnostic property of the Feature Lock Attack.

**Logistic Regression (LR)** is a linear classifier trained with L-BFGS optimization. It acts as an interpretable baseline. The linear structure of logistic regression means that the outputs produced by the permutation-based SHAP estimator can directly verify the theoretical guarantee.

**Neural Network (NN − TabNet)** is a feedforward network with a dropout of 0.3. We use `BatchNorm1d` for training stability on small tabular datasets. Furthermore, the training uses a `ReduceLROnPlateau` scheduler (patience = 10, factor = 0.5) and Adam with weight decay $10^{-4}$.

## A.3  Evaluation Metrics

**Accuracy Gain.** We report the accuracy gain to quantify the accuracy gain for using the protected feature $x_p$:

$$\Delta_{\text{acc}} = \text{acc}(f_P) - \text{acc}(f_{NP}). \tag{18}$$

**Baseline Attribution.** The mean absolute attribution of the protected feature $x_p$ before any adversarial attack.

**Attacked Attribution.** The mean absolute attribution of the protected feature $x_p$ post the Feature Lock Attack.

**Attribution Reduction.** The percentage of attribution of the protected feature $x_p$ suppressed post the Feature Lock Attack.

**Feature Weight Amplification Curve.** We use it to visualize how the baseline attribution scales with the increase in value of $\alpha$, while the attacked attribution remains unchanged.

## A.4  Explainer Configuration

**SHAP.** We deploy the permutation-based SHAP explainer throughout our experiments. This choice is necessary to ensure cross-model compatibility and to avoid model-specific inductive biases.

**LIME.** Throughout our experiments, LIME is configured with `discretize_continuous = False`, as with discretization enabled, binary features are binned into the same bin, which creates zero variation regardless of the model. When the discretization is disabled, LIME takes a sample of perturbed values from a Gaussian

distribution around the instance value. This leads to producing a variation that the linear coefficient of LIME captures.

