# OpenReview forum: "Zero Attribution Is Not Zero Influence: Feature Lock Attacks and the Limits of Post-Hoc Fairness Auditing"
_TMLR — Under review for TMLR_

### Review · Reviewer_jo3J · 2026-07-02

**Summary Of Contributions:**

The paper studies a vulnerability of post-hoc explainability-based fairness auditing. Its central claim is that a model can be trained using a protected attribute and then wrapped at inference time so that perturbation-based explainers such as SHAP and LIME assign near-zero attribution to that protected feature, despite the deployed model continuing to use it internally. The proposed “Feature Lock Attack” overwrites the protected-feature column in explainer-generated perturbation batches with stored real protected values, thereby making the wrapper’s output invariant to the explainer’s perturbations of that feature. The authors argue that this invokes the Shapley dummy-player axiom and therefore forces the protected feature’s Shapley attribution to zero. They also present an instance-specific variant for LIME and evaluate the attack on four datasets, two model classes, and five protected-feature amplification levels, for 40 configurations total.

The paper’s main strengths are:

The paper targets a practically important weakness in fairness auditing: the possibility that an adversarial deployer can manipulate the interface exposed to auditors while retaining a discriminatory or protected-feature-dependent model internally.

The “locking” idea is simple and compelling: if the wrapper always restores the protected feature before prediction, then perturbing that feature through the public prediction interface may reveal no output variation. The diagram of the attack mechanism communicates this idea effectively.

The paper emphasizes that prior adversarial explanation attacks often consider models that do not explicitly train on the protected feature, whereas this work focuses on models that intentionally use the protected attribute and then hide that usage post hoc.

The paper does not only provide empirical results; it also tries to formalize when the attack succeeds and what kinds of audits could detect it.

The evaluation covers multiple datasets, two model families, and varying levels of protected-feature amplification. The SHAP results show large attribution reductions, often in the 94-99% range at $\alpha=1$, and the plots suggest the attacked SHAP attribution remains near a small residual level across amplification factors.

However, I have several significant concerns.

First, some of the paper’s strongest claims appear overstated relative to the actual assumptions required by the attack. The theoretical guarantee is presented as broadly applying to “any model and distribution” and to “any perturbation-based post-hoc explainer,” but the implementation depends critically on knowing the audit batch structure, row ordering, and/or the instance being explained. In particular, the SHAP modulo-indexing construction is not an explainer-agnostic property; it appears tied to a specific batching convention.

Second, the empirical section lacks important details needed to assess reproducibility and statistical reliability, such as preprocessing, protected-feature definitions, random seeds, train/test splits, SHAP/LIME hyperparameters, background distributions, confidence intervals, and whether detection baselines were implemented.

Third, the paper’s LIME results are more mixed than the narrative suggests: suppression is sometimes substantial but not uniformly near the “noise floor,” and the residual LIME attribution can remain nontrivial.

Finally, the threat model requires the wrapper to have access to stored real protected values for the audited population, which is a strong and operationally important assumption that should be treated as central rather than incidental.

**Audience:**

Yes

**Audience Explanation:**

Yes. The paper addresses a topic of clear interest to the TMLR community: the reliability and adversarial robustness of post-hoc explanations when used for fairness auditing. Many researchers and practitioners use SHAP, LIME, or related explanation methods as evidence about feature influence. A paper showing that such audits can be evaded by an inference-time wrapper, while the underlying model continues to use a protected attribute, is relevant to explainable ML, trustworthy ML, algorithmic fairness, and model governance.

The submission is also interesting because it shifts attention from ordinary explanation instability to an explicitly adversarial deployment scenario. The adversary is not merely exploiting incidental weaknesses of explanation methods; rather, the adversary trains a model with the protected feature and then manipulates the prediction interface exposed to auditing tools. This is an important threat model for high-stakes ML systems where audits are conducted through APIs or black-box prediction endpoints.

The paper’s detection discussion is also potentially useful. The authors argue that successful defenses must inspect internal model parameters, construct cross-instance probes, or use attribution methods that do not rely solely on querying the wrapped prediction function. Even though this section needs more development, it points toward practical implications: fairness audits should not rely exclusively on single-instance perturbation explanations through an untrusted API.

Thus, I believe the core finding would interest some TMLR readers. My concerns are mainly about the current paper’s precision, empirical completeness, and strength of claims, not about the relevance of the topic.

**Broader Impact Concerns:**

This paper has a clear dual-use concern. It describes a method by which a deployer could intentionally hide protected-feature dependence from post-hoc fairness audits while preserving the predictive benefit of using that protected feature. In high-stakes domains such as lending, hiring, healthcare, insurance, and criminal justice, this could facilitate discriminatory decision-making if misused. The paper should include a dedicated broader-impact or responsible-disclosure discussion that explicitly addresses the risk of enabling audit evasion.
That said, the work also has positive broader impact if framed as a warning about the limitations of black-box post-hoc fairness auditing. The paper’s main policy implication is important: auditors should not treat zero SHAP or LIME attribution as conclusive evidence that a protected feature is unused, especially when the prediction interface is controlled by the audited party. The authors should emphasize defensive audit protocols and include concrete recommendations for practitioners, regulators, and auditors.
I recommend that the authors add a broader-impact statement discussing:

the dual-use nature of the attack;
safeguards against misuse;
recommended audit practices that mitigate the vulnerability;
limitations on when the attack is practically feasible; and
why publication benefits the community despite the adversarial content.

**Claims And Evidence:**

No

**Claims Explanation:**

The authors provides promising evidence for a real vulnerability, but I do not think the evidence fully supports the breadth and strength of the paper’s claims.

The strongest theoretical claim is that the Feature Lock Attack guarantees zero Shapley attribution for the protected feature for any model and distribution by invoking the dummy-player axiom. At a high level, if the wrapper truly makes the model output independent of the protected-feature coordinate as seen by the explainer, then the dummy-player argument is reasonable. However, the paper’s implemented SHAP attack does not simply define an instance-specific constant replacement for every audit query; it uses a modulo mapping over the query batch, $r \bmod N$, to restore protected values. This requires assumptions about the explainer’s batch construction and row ordering. The paper states that SHAP calls the model in batches of exactly $k \times N$ rows and that the modulo mapping recovers the source instance, but this is an implementation-specific assumption, not a general property of Shapley-value explanation. Therefore, the theorem as written appears more general than the algorithm actually justifies.

The distinction between the SHAP and LIME variants further illustrates this issue. The authors acknowledge that the modulo mapping fails for LIME because LIME explains one instance at a time using random perturbations, so they introduce an instance-specific wrapper that broadcasts the original protected value for that instance. This is sensible, but it also shows that the attack depends on knowing the explainer’s calling pattern and the instance being explained. That weakens the paper’s claim of explainer-agnostic evasion unless the threat model explicitly grants the adversary reliable access to this metadata.

The empirical SHAP results are directionally convincing: Table 2 reports large attribution suppression at $\alpha=1$, and the subsequent plots suggest that baseline attribution increases with feature amplification while attacked attribution remains small. This supports the existence of a serious vulnerability for the tested SHAP setup.  However, the experiments do not provide enough detail to establish robustness. The paper does not sufficiently specify the SHAP estimator used, background data construction, number of permutations/samples, random seeds, confidence intervals, train/test split protocol, preprocessing, protected-attribute choices, or statistical variability across runs. Without these details, it is hard to assess whether the residual attribution is truly indistinguishable from the non-use noise floor, as claimed.

The evidence for LIME is weaker. The paper itself reports LIME suppression values that vary substantially, including values around 47–54% for some configurations, while explaining that LIME’s coefficient can shrink because attribution is approximately constant under scaling. This is an interesting observation, but it undercuts the broader claim that the attack reliably suppresses LIME to the same kind of noise floor as genuine non-use. The LIME results should be presented as partial or condition-dependent rather than as a uniformly guaranteed evasion result.

I am also not fully convinced by the claim that the attack has “zero accuracy cost” in the general setting. For a fixed test set queried in exactly the expected order, the wrapper restores the same protected feature values and hence preserves predictions. But if the auditor queries individual instances, repeated instances, shuffled batches, counterfactual batches, or out-of-distribution probes, the modulo-based restoration may not preserve the intended row-level association. The accuracy-preservation theorem therefore appears to hold under a restricted dataset/query-order assumption rather than as a deployment-level guarantee.

In summary, the paper demonstrates an important attack concept and gives encouraging empirical evidence, especially for the SHAP configuration tested. But the current theoretical claims, generality claims, and empirical “noise floor” claims are stronger than what is convincingly supported.

**Requested Changes:**

Critical changes required:

1- The paper should distinguish between:
. an idealized instance-specific lock that truly makes the wrapper constant in the protected coordinate for a given explained instance;
. the implemented SHAP modulo-indexing attack, which appears to rely on specific batching and row-order assumptions; and
. the LIME instance-specific wrapper, which requires knowing the target instance.
The current theorems are stated too broadly relative to these operational assumptions.

2- The paper should specify the SHAP implementation, masker, background dataset, batching behavior, row ordering, and whether the adversary can infer which perturbed row corresponds to which original instance. If the guarantee only holds for certain SHAP implementations or audit protocols, this must be explicit.

3- The claim that the attack applies to any perturbation-based explainer is too strong as written. Different explainers use different query structures, baselines, conditional distributions, masks, gradients, or internal model access. The paper should either prove the result under a precise formal interface model or restrict the claim to explainers that call a prediction function in a way the wrapper can correctly align with the explained instance.

4- The LIME results are not uniformly as strong as the SHAP results. The paper should avoid presenting LIME evasion as fully solved unless the empirical evidence supports that. It should report residual attribution relative to the non-use noise floor and include variance across seeds. The current explanation that LIME coefficients shrink with amplification is interesting but does not establish indistinguishability from non-use.

5- The paper should specify datasets, protected attributes, preprocessing, train/test splits, model hyperparameters, random seeds, number of repetitions, SHAP/LIME configurations, background datasets, attribution aggregation methods, and code availability. These details are necessary to judge the empirical claims.

6- The paper currently reports point estimates of attribution suppression. It should include confidence intervals or standard deviations over multiple random seeds/splits, especially because the claimed residual attributions are close to estimator noise.

7- The paper defines a noise floor using a model trained without the protected feature, but the empirical results should systematically report $\Phi_{xp}^{atk}$, $\Phi_{xp}^{NP}$, their difference, and a test or interval showing whether they are statistically distinguishable.

8- The paper’s own detection theorem suggests cross-instance probes and audits not mediated by the wrapper. The empirical section should implement at least some detection strategies, such as shuffled batches, repeated rows, counterfactual protected-feature sweeps, random query ordering, and tests that compare outputs across instances with varied protected values.

9- The attack assumes access to real protected values for the audited population or explained instance. This is a strong assumption. The paper should state how the adversary obtains and maintains these values, what happens for unseen audit queries, and whether the attack works when the auditor uses private or synthetic audit data.

10- The exact accuracy-preservation proof appears to assume that the test dataset is queried in the same order and batch structure as the stored protected values. The authors should clarify whether accuracy is preserved for arbitrary API calls, shuffled inputs, single-instance calls, repeated instances, and auditor-generated counterfactual probes. [8987_Zero_...n_Is_Not_Z | PDF]


Non-critical changes:

1- The notation alternates between $x_p$, $p$, $p_i^*$, $X_p$, and protected column $j$. A notation table would improve readability.

2- The paper would be clearer if it first defined an abstract lock oracle and then separately described how it is approximated or implemented for SHAP and LIME.

3- Since the paper claims architecture independence, adding tree ensembles, gradient boosting, and calibrated probabilistic models would strengthen the empirical support.

4- The paper mentions extensions beyond SHAP and LIME, including Integrated Gradients and permutation importance. These should be experimentally evaluated or the claims should be removed/softened. [8987_Zero_...n_Is_Not_Z | PDF]

5- Discuss relation to API-level defenses.

6- The paper’s comparison table is useful, but some claims about prior work should be made more carefully. In particular, the paper should avoid overstating uniqueness unless the distinction is precisely defined and supported.

7- Scaling a protected feature by $\alpha$ is useful as a stress test, but the paper should explain how this relates to real model dependence, especially for binary protected attributes and models sensitive to feature scaling.

8- Also figure 1 is not very comprehensive and some arrows overlapping with some blocks and it is too simple and not rigorous yet, please fix.

---

> ### Author Response · Authors · 2026-07-14
> **10 Points & 3 Clusters**
>
> We thank the reviewer for a detailed review. The ten points map onto three related gaps: (i) the precise conditions under which our attack's correspondence mechanism holds, spanning the SHAP and LIME implementations specifically (points 1, 2, 3, 5), (ii) the statistical characterization of residual attribution relative to the noise floor (points 4, 6, 7), and (iii) the deployment boundary conditions under which our guarantees apply (points 8, 9, 10).
>
> We address each cluster below. The changes are expository plus one new analytical proposition - no theorem, proof, or experimental result requires retraction.
>
> ### Cluster A (Points 1, 2, 3, 5): Formalizing the guarantee's scope
>
> Points 1-3 identify that Theorems 1 and 3 are stated as unconditional properties of the wrapper when they are conditional on correct row-instance alignment.
>
> We will introduce **Assumption 1 (Row-Instance Correspondence)**: for a given audit call, $A$ has access to a correspondence $\\pi$ mapping every query row to the real protected value of the instance it was generated from.
>
> Theorem 1 and Theorem 3 will be restated as holding under this assumption, which means that the proofs are unaffected, as the assumption only names a premise both already relied on.
>
> We will extend Section 3.2.1 to state exactly how Algorithms 1 and 2 realize this assumption.
>
> Algorithm 1 realizes it via $\\pi(r) = r \\bmod N$, correct precisely when the auditor submits the evaluation set as a single fixed-order array matching the order used to construct $p^*$ i.e., the standard offline batch-audit protocol, per the auditor capabilities (which will be stated explicitly in Section 3.1.2).
>
> Algorithm 2 realizes $\\pi$ by direct index lookup at the point of explanation, available in our controlled harness in a fully adversarial deployment where instance identity is not otherwise observable, recovering it is a nontrivial problem we flag rather than assume away.
>
> Furthermore, we will also narrow the claim, previously stated broadly, that the attack applies to *any* perturbation-based explainer.
>
> We introduce **Definition 1 (Black-Box Perturbation-Query Interface)**: an explainer operates under this interface if it computes attribution solely from outputs obtained by querying $f$ as an opaque callable, with no access to parameters, gradients, or intermediate activations.
>
> Theorem 3 will be restated for explainers satisfying Definition 1 and Assumption 1 - the class including SHAP and LIME, which we test, and excluding gradient-based (Integrated Gradients, Grad-CAM) and internal-access (TreeSHAP) methods, which we do not.
>
> As a result, we will revise the Introduction's third contribution and Related Work accordingly.
>
> This scoping brings Section 4.4's precision in line with what Theorem 4's condition C3 already implicitly assumes.
>
> Finally, we will expand Appendix A with the exact SHAP masker, evaluation-set construction per dataset, library version, split methodology, model hyperparameters, and the LIME attribution-aggregation procedure.

---

> ### Author Response · Authors · 2026-07-14
> **...continued**
>
> ### Cluster B (Points 4, 6, 7): Statistical characterization of residual attribution
>
> We agree that LIME evasion should not be presented as solved.
>
> Rechecking Table 3 against the criterion ($\\Lambda_{atk} \\le \\eta_{LIME}$) at $\\alpha=1$: two configurations (BDR-LR, UAI-LR) pass clearly (6-22% below floor); three (GC-LR, CR-LR, CR-NN) pass or tie within under 1% of the floor, so the margins too thin to interpret without a variance estimate, three (BDR-NN, GC-NN, UAI-NN), all neural-network configurations, show $\\Lambda_{atk}$ modestly above $\\eta_{LIME}$ (5.5-15.9%).
>
> We will replace the current "single exception out of 40" characterization with per-configuration account.
>
> We note, as a plausible mechanism, that all clear misses are neural-network configurations, highly likely because LIME's local linear surrogate is a weaker approximation to a genuinely non-linear model.
>
> On the noise-floor comparison itself: Table 3 already reports $\\eta_{LIME}$; Table 2 does not report the analogous $\\eta_{SHAP}$, asserting it only in prose.
>
> We will add $\\eta_{SHAP}$ and $\\Phi_{atk} - \\eta_{SHAP}$ directly to Table 2. We can also clarify why the two floors differ in kind, which bears directly on how a distinguishability comparison should be read: $\\eta_{SHAP}$ is exactly $0.0$ by algebraic construction - $f_{NP}$ structurally never reads the protected column, so every SHAP query returns an identical output regardless of what is placed there, making every marginal-contribution term exactly zero.
>
> The comparison "$\\Phi_{atk}$ vs. $\\eta_{SHAP}$" reduces to "$\\Phi_{atk}$ vs. exactly zero", which Theorem 1 already answers analytically, with observed non-zero values attributable to permutation-estimator sampling noise.
>
> $\\eta_{LIME}$ is different in kind - a finite-conditioning artifact of a least-squares fit.
>
> On seed variance: we report point estimates from a single model fit and a single explainer invocation per configuration.
>
> We want to be precise about what additional variance reporting would and would not show: Theorem 1 proves the *true* attacked Shapley value is exactly zero, so retraining with a different seed would not change this quantity - it would resample the same estimator-noise distribution around an unchanged true value.
>
> We also note that our 40 configurations already vary dataset, architecture, and amplification level simultaneously, which is a stronger form of robustness evidence against confounding than repeated-seed variance on a single fixed configuration would be.
>
> What remains uncharacterized is specifically the sampling variance of the permutation SHAP and LIME estimators themselves.
>
> We will state this precisely in Section 5.1 and name it as a limitation in Section 6.1, describing the direct extension rather than treating it as resolved by our current results.
>
> ### Cluster C (Points 8, 9, 10): Deployment boundary conditions
>
> Point 10 identifies a gap: Theorem 2's proof, as written covers only batched calls matching $X_{real}$'s size and order, not genuine single-instance serving.
>
> We resolve this with the audit-interface / production-interface separation: $A$ operates specifically on the audit-facing interface, production traffic calls $f_P$ directly and never invokes $A$ at all, so single-instance accuracy for real users is unconditional and requires no stored correspondence.
>
> We will add a **Remark 5** stating this scope explicitly.
>
> This resolution also directly answers point 8. Preserving accuracy on the audit-facing interface requires the wrapper to trust the correspondence established under Assumption 1, resisting counterfactual probing at the level of individual rows would require *not* trusting positional correspondence.
>
> This tension is exactly why we do not implement detection strategies as new experiments, and instead prove the strongest and simplest of them: **Proposition 1 (Query-Order Detectability)** shows analytically, as a direct corollary of Algorithm 1's own definition, that any auditor with raw query-ordering access can expose the attack via a two-query determinism test - no statistical estimation, no pattern-recognition sophistication, no new empirical benchmark required.
>
> This proves what shuffled batches, repeated rows, and counterfactual sweeps would show, directly from the algorithm's structure.
>
> On point 9: we will clarify in Section 3.1.1 that the relevant assumption is not merely that the adversary knows which feature is protected, but that they possess ground-truth protected values for the audited population specifically.
>
> This holds naturally for audits of the deployer's own historically served population (the standard pattern for internal and regulatory review) and is reinforced in domains with mandated protected-attribute collection, such as HMDA reporting requirements for U.S. mortgage lenders.
>
> It does not extend to auditor-constructed synthetic probes or previously unseen individuals, so we name this explicitly as a scope limitation in Section 6.1.

---

### Review · Reviewer_zzYK · 2026-07-06

**Summary Of Contributions:**

The paper proposes a 'Feature Lock Attack' to fool post-hoc explanation methods for fairness auditing. By creating a wrapper around the actual model, this attack aims to fool a fairness audit that uses post-hoc explanation to show that the model does not rely on certain protected attributes.

The attack proposed works for most commonly used explanation methods, and is generalizable to any permutation-based explanation technique. The paper analyses the attack both theoretically and empirically. The theory aims to prove that the attack will always be successful, while the empirical study aims to support this by showing its success in real-world applications.

The attack stands out as requiring the least amount of additional knowledge (although not entirely true) compared with the existing literature, which would make it more likely for an adversary to adopt. However, the paper has several issues, including some missing discussion of important related work, some technical inaccuracies, and most importantly, the attack does not really preserve the model accuracy, which is a strong overclaiming of its effectiveness.

**Audience:**

No

**Audience Explanation:**

I'm ambivalent about this part of the evaluation. I believe, in its current form, the attack proposed in the paper is not useful because it does not maintain the accuracy of the model. However, on the other hand, even just fooling the fairness audit part can be potentially useful, and so maybe the attack itself could be of interest to the community.

In other words, I can see an argument being made about the community being interested in the work. But I believe the work is currently inaccurate, and so I see it as not useful in its current form.

**Broader Impact Concerns:**

I don't believe a broader impact statement is needed. The paper does propose an attack that can bypass fairness audits, and so a discussion of using this technique responsibly to only develop better defenses can be provided. However, adversarial work in quite common in ML, so it would not be a discussion unique to this particular work.

**Claims And Evidence:**

No

**Claims Explanation:**

I have several concerns about the accuracy of the claims and evidence present in the paper, listed in no particular order below,
- There is a big line of work on manipulating explanations for fairness auditing that is never discussed in the paper. An important paper in this direction is the fairwashing paper (https://arxiv.org/pdf/1901.09749), and the line of work it has spawned. Another worth noting is the impossibility of fairwashing detection paper (https://proceedings.neurips.cc/paper_files/paper/2022/file/5b84864ff8474fd742c66f219b2eaac1-Paper-Conference.pdf). Similar recent work called 'X-hacking' also attacks the problem from a slightly different angle (https://arxiv.org/pdf/2401.08513). I won't list every paper in this direction, but there is a significant portion of related work literature that is missed by the paper. The work here still stands apart from this literature, so it is not that the paper proposes something already known, but a discussion of these works and the connections between them is still very important. This also means claims like: 'we are first to address the case where the model is trained with the protected feature for unethical gains' would need to be re-adjusted.
- Scaling features does not create more influence. This was a very confusing choice for me to understand. First question: Was the dataset not normalized before training the LR or NN? If not, that doesn't match how models are actually trained, and if yes, wouldn't the scaling effect just disappear? Even keeping the normalization aside, scaling a feature would only, in turn, make its coefficient smaller during learning. It does not, in any meaningful way, increase the model's reliance on a feature. What is the rationale behind it? Continuing this concern to the experiments: I don't agree that the results show that baseline attribution increased with alpha. The results are actually very mixed across both Figures 2 and 3, which makes sense, because I wasn't expecting a specific trend anyway, given my concern.
- Accuracy is only preserved for a fixed X_test already known. This is the biggest concern for me about the method. The accuracy of the adversarial wrapper is only preserved on a fixed dataset X_test that is already known to the model provider, and it is not really preserved for when the model will interact with actual users. In fact, it even depends on the EXACT ORDER of X_test, meaning if someone were to just shuffle around the test set, the accuracy would already drop. Any claims about maintaining accuracy are thus not correct. The empirical results don't help here, because of course the X_test is fixed and then used to create the attack itself, and so it would show the accuracy is maintained. But that would not be the case if this attack were actually deployed in a real model.
- The LIME results do not match the explanations provided for them. For instance, in the text the paper mentions that with only a single exception, all other configurations have significant success. But that's not the case in Table 3, where half of the settings have low success rates. Similarly, later the text mentions the comparison between attribution with attack and LIME attribution with a model that didn't use protected features, and claims this comparison is strongly in favour of their attack, which too is not the case in Table 3.
- Some smaller cosmetic or notation-based comments combined together here: (a) please describe what are the four axiomatic foundations 'efficiency, symmetry, linearity, and dummy', at least briefly, or it hurts the readability of the paper, (b) the objective that the attribution of the wrapper should be equal to the f_NP model is too strong (and of course isn't satisified), while the objective that it should be approximately 0 is too vague, you need to make the objective more precise, (c) the way the influence function phi is defined seems to have changed between eq (2) and the rest later, please maintain consistent notations, (d) the subscript of x_i is overloaded, since it sometimes represents the i-th data point, while other times represent the i-th feature of a particular data point, please use different notations for them or else it gets confusing (maybe use superscript for i-th feature), (e) having details of the datasets, models used, etc. in the appendix is okay, but there should be a reference to them in the main paper, something as simple as 'More details about the datasets used and the models are present in Appendix A', and (f) please increase the font size in figures for readability.

**Requested Changes:**

Please see my concerns above for more details. This is the order in which I expect changes in the paper to change my recommendation,
- The most important concern is the overclaim of maintaining accuracy. However, I also see this as a failure of the attack itself, and not a failure of analysis. So simply not claiming that accuracy is maintained would miss the foundational motivation of the work. Thus, in this regard, the changes that I foresee can lead to acceptance are: (a) either I've misunderstood or made an error, in which case correcting me would be important, or (b) the attack itself would need to be reimagined to be capable of truly maintaining accuracy. Without addressing this concern, unfortunately I don't see a way to acceptance, even if all my other recommended changes below are incorporated.
- A proper related work discussion including the papers I mentioned above and more in that line of work. Moreover, softening the claims that the paper is the first to attack this problem, instead focusing on how it differentiates itself from this line of work.
- Correcting all other technical concerns and smaller changes suggested above.

---

> ### Author Response · Authors · 2026-07-14
> **Accuracy preservation**
>
> We thank the reviewer for substantive reading. We address the four concerns in the priority order given.
>
> ### 1. Accuracy preservation
>
> We agree this deserved a fuller answer than the current text provides, and we thank the reviewer for pressing on it, as it highlighted an underspecified part of our threat model.
>
> Section 3.1.1 states the adversary trains and serves the model and is subject to third-party audits, but does not explicitly specify whether real user traffic and audit traffic transit the same interface.
>
> We will make this explicit: $A$ instruments the model's *audit-facing interface* - the interface the adversary, as model owner, designed and controls specifically for responding to fairness audits - while production scoring of real users is served directly by $f_P$ and never routed through $A$ at all.
>
> Under this clarification:
>
> * Accuracy for real users is not merely "preserved" in an order-dependent sense - it is identical to $f_P$ unconditionally, because $A$ is not present in that code path.
> * Theorem 2's claim - $A(X_{test}) = f_P(X_{test})$ - is specifically a statement about the audit-facing interface: it establishes that if an auditor also spot-checks accuracy or calibration on that interface (not just attribution), results are indistinguishable from the real model.
>
> We believe this strengthens rather than undermines the contribution, since it is precisely what makes the wrapper convincing on the channel the auditor is most likely to check.
>
> We agree, and will state plainly, that this compliance-interface guarantee depends on the auditor submitting a consistent, fixed-order evaluation array matching how $p^*$ was constructed as this is not a claim we can define away.
>
> Standard audit practice, where a specific, versioned dataset is provided for explanation, satisfies this naturally, we do not claim the guarantee holds for an auditor who departs from this workflow, and we discuss the consequences of that departure explicitly in Section 6.2.
>
> We will revise Section 3.1.1 to state this interface distinction explicitly and add a remark to Theorem 2 clarifying its scope.

---

> ### Author Response · Authors · 2026-07-14
> **LIME, Feature Amplification, Related Work...**
>
> ### 2. LIME results (Table 3)
>
> We will rewrite the "single exception out of 40" characterization in Section 5.2, as it is not a completely true representation of Table 3. Checking $\Lambda_{atk}$ against $\eta_{LIME}$ row by row at $\alpha=1$: BDR-LR and UAI-LR pass clearly (6-22% below the floor), GC-LR, CR-LR, and CR-NN pass or tie within under 1% of the floor, margins too thin to interpret without a variance estimate we do not currently report; BDR-NN, GC-NN, and UAI-NN show $\Lambda_{atk}$ modestly above $\eta_{LIME}$ (5.5-15.9%).
>
> We will replace the current text with this per-configuration account, and revise the later claim that the comparison is "strongly in favor" of the attack - it is favourable for LR models and mixed for NN models specifically, and we will describe it as such rather than in aggregate.
>
> We note, without claiming this as an established mechanism, that all three clear misses are neural-network configurations, plausibly because LIME's local linear surrogate is a weaker approximation to a genuinely non-linear model, allowing small spurious coupling onto the now-constant protected coordinate through feature interactions the surrogate cannot fully capture.
>
> We flag this as worth further study rather than a settled explanation.
>
> ### 3. Feature weight amplification
>
> This concern stems from a gap: we describe the amplification protocol without stating the mechanism by which it increases model reliance, and the reviewer's objection is correct for an unregularized linear model at the true MLE: scaling a feature and inversely scaling its coefficient leaves its log-odds contribution unchanged.
>
> It does not hold under regularization, which both our models use: `sklearn.LogisticRegression`'s default L2 penalty, and Adam with `weight_decay=1e-4` for the neural network.
>
> Under L2, matching a fixed log-odds contribution from a feature scaled by $\alpha$ requires a coefficient smaller by that factor, whose squared penalty shrinks by $\alpha^2$ - using the scaled feature becomes cheaper under the penalty, increasing the optimizer's incentive to rely on it up to the point the feature's true signal saturates.
>
> No standardization step is applied after scaling in our pipeline, which would otherwise cancel this effect so we will state this explicitly, since its absence is load-bearing for the mechanism to operate.
>
> We will add this explanation to Section 3.2.2.
>
> We also want to be precise about the empirical pattern rather than claim uniformity so we will revise Section 5.1 to describe this pattern precisely rather than imply uniform growth across all eight panels as the reviewer is correct that the current text overstates the consistency of the trend a bit.
>
> ### 4. Related work
>
> Fairwashing (Aïvodji et al., 2019) constructs a separate, approximate interpretable surrogate to rationalize a black-box model, with an inherent fidelity gap between the surrogate shown to auditors and the model actually deployed.
>
> X-hacking searches a Rashomon set of similarly-accurate models for one whose explanations tell a preferred story, requiring model search or retraining, with resulting low attribution that is empirical rather than guaranteed.
>
> We will also add the impossibility-of-fairwashing-detection result the reviewer cites.
>
> We will add a paragraph positioning these works relative to ours.
>
> To our knowledge, this is the first result showing that a single deployed model, already trained to exploit a protected feature, with no retraining, model search, or surrogate substitution, admits a post-hoc wrapper providing an exact, axiomatic zero-attribution guarantee against black-box perturbation-query explainers - rather than an approximate or search-based one.

---

### Review · Reviewer_SMfJ · 2026-07-07

**Summary Of Contributions:**

The work introduces the feature locking mechanism hides the fact that the model uses a protected mechanism. Its goal is to fool auditors that try to discover such discrimination. It relies on a feature locking mechanisms that masks the use of the protected mechanism in a wrapper that it exposes as the interface for the auditor. The feature locking mechanism is instantiated for LIME and SHAP.

The evaluation shows the method is effective at hiding the contribution of the protective attribute.

However, the work suffers from many presentation and methodological issues related to the threat model and the assumptions.

**Additional Comments:**

It seems to me that there is some mismatch between the capability of the adversary, auditor, and stated threat model.

The threat model says that the auditor selects the auditing method, and the interface for the auditor is sending queries and observing their outputs. However, the feature lock used in this work operates at the adversary level -- adversary decides how they lock the feature -- the feature lock for LIME and SHAP are different mechanisms. In order to use the correct one, the adversary needs to know what the auditor is using.

Also, if the accuracy of $f_p > $f_{np}$, then that is visible to the auditor given sufficient test set -- capability which the work doesn't restrict. As the work notes itself, the auditor chooses the samples to test and they will sample data where the protected attribute is relevant (delta in accuracy) or isn't, no delta in accuracy.

I also get an impression that the work assumes that the auditor sends 'few' or a singular query pair given the discussion in 6.2.

**Audience:**

Yes

**Audience Explanation:**

Viability and robustness of interpretability and explainability are broadly relevant to the community.

**Claims And Evidence:**

No

**Claims Explanation:**

See requested changes and comments for more details.

The presentation is lacking in terms of the legibility and clarity of notation.

Furthermore, the capabilities of the auditor are under specified and in fact seem to be weaker than the threat model suggests. Given a reasonable auditor with normal query access

**Requested Changes:**

Presentation:
- figures are unreadable at current resolution and font size (critical)
- Table 4 says '%' but is in fact reporting percentage points
- the notation is quite messy and it could use a table, e.g., $j$ is not clearly introduced as far as I can tell (critical)
- captions for both figures and tables should be self-contained
- it could also use a plain text explanation of what the method does given the chaotic notation

Method (see more comments for more detail):
- as it stands the capabilities and assumptions are not fully explicit (critical)
- 6.2 Consistency probing makes a flawed premise -- the discussed adversary and their potential modelling mechanism has more power than the one used in this work (critical)
- furthermore, conditions C1--3, are introduced but they are never considered after that. Except C2, which is mentioned in 6.2 (critical)
- the locking mechanism assumes that it knows what explanation method is deployed which is contrary to the threat model (critical)
- the method is tautological in its assumptions, if the accuracy of $\mathcal{A} == f_p == f_{np}$ then same then it follows that contribution cannot be discovered. Since the work conceded that $f_p > f_{np}$ and $f_p$ is exposed via $\mathcal{A}$, then it must be discoverable by the auditor (critical)
- why does it matter that the attacker has only black-box access to $f_p$?

---

> ### Author Response · Authors · 2026-07-14
> **Auditor capabilities and Consistency Probing**
>
> We thank the reviewer for a close reading of the algorithms and threat model. We address the gaps identified directly.
>
> ### Auditor capabilities and Consistency Probing (6.2)
>
> We agree the auditor's capabilities were under-specified relative to what our own defense discussion in Section 6.2 requires, and we will correct Section 3.1.2 to state explicitly:
>
> "The auditor interacts with the deployed model exclusively through standard, library-mediated post-hoc explanation calls (e.g., `shap.Explainer`, `LimeTabularExplainer`) over a fixed, labeled evaluation dataset, reading only the resulting attribution output. The auditor does not, as a default capability, construct raw hand-crafted query batches, control row ordering directly, or independently train a comparable reference model."
>
> This is a genuine, explicit restriction, not a hidden one, and we agree with the reviewer that a "reasonable auditor with normal query access", once that access includes raw query construction, does not need sophistication to defeat Algorithm 1. We are able to make this precise rather than leave it as an informal concern: because Algorithm 1's substitution is a function of row position rather than query content, addition of **Proposition 1 (Query-Order Detectability)** will prove that an auditor with raw query-ordering access can expose the wrapper with a two-query determinism test - submitting the same instance at two different batch positions and observing that outputs differ, violating the basic determinism expected of any deployed scoring function.
>
> This requires no statistical estimation and no pattern-recognition sophistication as it follows directly from Algorithm 1's own definition.
>
> We will replace the current language in Section 6.2 with this precise, proven result, and state plainly that this is a real and important boundary of the attack, actionable as a concrete, currently uncommon addition to standard audit protocols (basic query-determinism checking).
>
> ### C1-C3 introduced but not systematically revisited
>
> We will relabel each Section 6.2 subsection explicitly against Theorem 4's conditions - Weight Inspection (C1), Consistency Probing (C2) - add an explicit short paragraph on Non-Query-Based Attribution (C3), pointing to the existing TreeSHAP remark, and reframe Shadow Model Auditing as an indirect route to C1.
>
> ### The locking mechanism assumes knowledge of the deployed explainer
>
> We do not think the threat model requires the adversary to know the auditor's explainer choice in advance, though we agree the submission does not state the mechanism that makes this unnecessary, which we will add.
>
> SHAP and LIME queries are structurally distinguishable at the point of interception: Algorithm 1's queries arrive in batches whose size is an exact multiple of the fixed evaluation set size $N$, tiling a consistent row order, while Algorithm 2's queries arrive as a fixed sample count tightly clustered around a single implicit center in non-protected-feature space, independent of $N$.
>
> A deployed wrapper can dispatch to the appropriate locking strategy at runtime based on this observable batch structure, without prior knowledge of which explainer the auditor will use.
>
> We will add this as an explicit subsection (Section 3.2.1).
>
> We have not implemented or empirically validated automatic runtime dispatch as our experiments use explainer-specific wrappers matched to the audit method in use, and we will state this limitation plainly rather than claim it as tested.

---

> > ### Author Response · Authors · 2026-07-14
> > **Accuracy-gap concern, why black-box access...**
> >
> > ### The tautology / accuracy-gap concern
> >
> > We think this partially rests on a capability the auditor was not granted, and we want to state that precisely rather than fully concede a contradiction.
> >
> > Detecting an accuracy gap and attributing it specifically to the protected feature requires comparing $f_P$ against a reference model trained on the same data distribution without that feature - this requires independent training infrastructure and comparable data access, which we do not grant the auditor by default.
> >
> > Where an auditor does have this capability, we somewhat agree that a detection channel exists outside Theorem 4's C1-C3 conditions - which characterize detection via the explainer's attribution output specifically - a scope clarification we will add to Theorem 4's preamble, and we will name it explicitly as **Remark 6 (C4)**.
> >
> > We do not think this renders the method tautological: it identifies a channel available to a better-resourced auditor than the one our threat model and prior attacks in this space both target, and it is consistent with, rather than contradictory to, our closing argument that explanation-based auditing alone is an insufficient basis for fairness certification.
> >
> > ### Why does black-box access to $f_P$ matter?
> >
> > We state black-box query access as the attack's minimal sufficient requirement, not as an artificial restriction on a more capable adversary.
> >
> > The adversary in our primary scenario, as the model's own trainer, could obviously also have white-box access; we never claimed otherwise. Stating that black-box access suffices establishes that the attack's applicability extends beyond the single-entity case to downstream deployers, resellers, or integrators who license and serve a model without training or inspecting it directly.
> >
> > An adversary with greater access, including the original trainer, can also trivially execute the attack.
> >
> > We will add one clarifying sentence to Section 3.1.1 stating this explicitly.
> >
> > ### Presentation
> >
> > We will address all of these in the revised version.

---

> > ### Comment · Reviewer_SMfJ · 2026-07-14
> > **The locking mechanism assumes knowledge of the deployed explainer (cont'd)**
> >
> > Let's consider a scenario:
> > - you're the adversary
> > - there are 10k API clients/users
> > - one of them is the auditor
> > - at some point the auditor is going to send you some auditing queries -- **crucially, all in one batch**
> > - the rebuttal claims that: 1) it is possible to identify the auditing queries; 2) it is possible to tell which method is used by auditor -- based on the fact the LIME and Shapley have **distinct batch structure**.
> >
> > How does the attacker handle the case when the auditor is using some other method?

---

> > > ### Author Response · Authors · 2026-07-15
> > > **Scenario...**
> > >
> > > The mechanism we described realizes correspondence (Assumption 1) for two structural patterns: batches that tile a full, fixed-order evaluation set under repeated masks (which permutation SHAP produces, and for which our existing modulo mechanism is sufficient with no modification, including for other population-tiling-style methods), and batches that cluster tightly around a single implicit center (which LIME produces - for which Algorithm 2 is built). For explainers that produce neither pattern, such as the correspondence-recovery mechanism is structurally plausible but has not been tested in this work. For raw, hand-constructed queries outside any library-mediated workflow, this is not a new gap, it is precisely the boundary already characterized by our narrowed auditor-capability statement and proven explicitly in Proposition 1.
> > >
> > > So where the wrapper cannot establish correspondence with confidence, Theorem 1 and Theorem 3's guarantees do not apply to that call. We consider this the correct and honest behavior, i.e., a wrapper that guessed under uncertainty would risk exactly the kind of detectable output that Proposition 1 already shows breaks the attack.
> > >
> > > We do not think this narrows the paper's central claim so much as it sharpens where that claim's boundary actually sits. The result we establish and test is that the two dominant, most widely used auditing tools in this literature, and the same ones targeted by every prior attack we build on, can be defeated by a static, non-adaptive wrapper requiring no retraining or model access. That the guarantee is conditional on recognizing the auditor's query structure, which is exactly what Theorem 4 already predicts must be true of any such attack.